# Diversity of Aβ aggregates produced in a gut-based *Drosophila* model of Alzheimer's disease

**Greta Elovsson**, **Therése Klingstedt, K. Peter R. Nilsson, Ann-Christin Brorsson***

Department of Physics, Chemistry, and Biology, Linköping University, Linköping, Sweden

* ann-christin.brorsson@liu.se

## Introduction

Alzheimer's disease (AD) is characterized by neuronal damage in the brain manifesting in memory loss and premature death. Unfortunately, an effective, disease-modifying treatment against AD remains to be discovered despite extensive research in the field. One major histopathological hallmark of AD is amyloid plaques composed of amyloid-β (Aβ) peptides [1]. Aβ is intrinsically disordered with a high tendency to undergo aggregation, which is a highly complex process resulting in various prefibrillar species and amyloid fibrils with different structures [2]. The Aβ polymorphism in AD introduces obstacles when it comes to structural determination, optical imaging, and possibly treatment. The aggregation process begins when Aβ misfolds, possessing a β-hairpin conformation, and then accumulates into larger species such as oligomers and protofibrils [3,4]. The aggregation continues until highly stable, insoluble amyloid fibrils – rich in β-sheet structure – are assembled [5]. The typical architecture of an Aβ fibril is protofilament(s) intertwined around each other in a spiral trajectory [6]. The protofilaments are long, fibrous structures made up of Aβ monomers stacked along the fibril axis, thus forming a repetitive β-sheet pattern called the cross-β motif [7–9]. In addition, Aβ fibrils exist in several polymorphs [2,10–13], with varying intertwining and width, which will affect the fibril's binding surfaces used by molecular probes for amyloid detection. Although Aβ fibrils may induce neuronal injury, evidence shows that soluble Aβ species, such as oligomers and protofibrils with a size of 80–500 kDa, greatly impact neurotoxicity and are considered pathogenic drivers [14–18]. Aβ oligomers can be produced via primary nucleation, where monomers are assembled, or by a catalytic pathway called secondary nucleation, where oligomeric species are effectively formed on the surface of fibrillar aggregates [19–21]. The secondary nucleation reaction promotes a positive feedback loop where prefibrillar species are continuously supplied [19]. Therefore, even though fibrils are considered inert, they provide binding surfaces for the secondary nucleation reaction to occur to generate neurotoxic oligomers.

Heterogeneity among Aβ oligomers and fibrils depends on the primary structure and the intrinsic aggregation property of the Aβ monomer. Aβ is derived from cleavage of the Aβ precursor protein (AβPP), where different Aβ isoforms, with varying lengths, are generated due to imprecise enzymatic cleavage [5,22–24]. The principal forms that are generated are Aβ1–40 and Aβ1–42, where the latter, with 42 amino

**Data availability statement:** All data files are available from the Dryad database https://doi.org/10.5061/dryad.b2rbnzssq.

**Funding:** Grant source: Vetenskapsrådet (Swedish research council) Specific grant numbers: 2016-00748 Initials of authors who received each award: K Peter R Nilsson Full names of commercial companies that funded the study or authors: This is not a commercial company URLs to sponsors' websites: English - Swedish Research Council The funders had no role in study design, data collection and analysis, decision to publish, or preparation of the manuscript. Grant source: Hjärnfonden Specific grant numbers: FO2020-0028 Initials of authors who received each award: K Peter R Nilsson Full names of commercial companies that funded the study or authors: This is not a commercial company URLs to sponsors' websites: Framåt! För hjärnans bästa, för allas skull - Hjärnfonden The funders had no role in study design, data collection and analysis, decision to publish, or preparation of the manuscript. Grant source: Hjärnfonden Specific grant numbers: FO2022-0072 Initials of authors who received each award: K Peter R Nilsson Full names of commercial companies that funded the study or authors: This is not a commercial company URLs to sponsors' websites: Framåt! För hjärnans bästa, för allas skull - Hjärnfonden The funders had no role in study design, data collection and analysis, decision to publish, or preparation of the manuscript. Grant source: Gun och Bertil Stohnes Stiftelse Specific grant numbers: N/A Initials of authors who received each award: Ann-Christin Brorsson Full names of commercial companies that funded the study or authors: This is not a commercial company URLs to sponsors' websites: O. E. och Edla Johanssons vetenskapliga stiftelse The funders had no role in study design, data collection and analysis, decision to publish, or preparation of the manuscript. Grant source: Åhlén-stiftelsen Specific grant numbers: 193059 Initials of authors who received each award: Ann-Christin Brorsson Full names of commercial companies that funded the study or authors: This is not a commercial company URLs to sponsors' websites: ÅHLÉN- STIFTELSEN The funders had no role in study design, data collection and analysis, decision to publish, or preparation of the manuscript. Grant source: Alzheimerfonden Specific grant numbers: AF-981115 Initials

acids, is very aggregation-prone. Aβ with the arctic mutation (E22G) is a familial variant associated with severe AD progression due to its accelerated aggregation. Quantitative analyses have revealed that Arctic Aβ promotes a rapid formation of protofibrils, associated with increased neurotoxicity [16,25,26]. Expression of Arctic Aβ in a *Drosophila* model has been shown to generate higher toxicity, manifested in a faster mortality rate and enhanced locomotor dysfunctions, compared to wildtype Aβ1–42 [27,28]. In 2012, a group of researchers synthesized and examined new Aβ variants to gain further insights into the mechanism behind Aβ aggregation and how it connects to neurotoxicity. By connecting two copies of Aβ1–42 (head-to-tail) with a linker and thus creating a dimeric tandem construct, high toxicity was achieved in *Drosophila* along with a massive amyloid load [29]. In addition, they reported that neurotoxicity correlated with the ability to populate stable and soluble oligomeric states, which is in line with previous studies [4,30]. The Aβ tandem construct with a linker displayed a more toxic behavior than wild-type Aβ1–42. The buildup of protein aggregates is widely regarded as a pathological trait of several neurodegenerative diseases, such as AD or Parkinson's disease, and for that reason, detection and characterization of aggregates are beneficial. Besides antibodies, fluorescent amyloid ligands such as luminescent conjugated oligothiophenes (LCOs) are valuable tools for optical imaging of different types of Aβ aggregates [31–34]. These light-emitting probes acquire a specific detectable signal upon binding cross-β-sheet structures, which are abundant in compact fibrillar Aβ. The inherent features of the ligand and how its thiophene-based backbone adopts to the target, decide the spectral properties of the fluorescent signal [35,36]. Consequently, the emission spectra generated by the amyloid ligand can be utilized to determine alternative binding modes of the ligand to distinct Aβ aggregates. LCOs HS-84 and HS-169 [37,38] have successfully labeled Aβ plaques in an AD mouse model and interestingly they do not seem to completely co-localize indicating structural differences between the targets [34]. The designs of HS-84 and HS-169 are rather similar apart from the centered compound. HS-169 has a central benzothiadiazole unit resulting in a stiffer, less flexible construction relative to HS-84 which instead has a central thiophene moiety [38].

The fruit fly, *Drosophila melanogaster*, has proved to be a highly useful model organism in biomedical sciences. By exploiting the Gal4/UAS system a target protein can be expressed in a specific tissue or cell type, thus making it possible to study pathological mechanisms behind human diseases [39].

In this study, we aimed to investigate the diversity of Aβ aggregates produced by flies expressing the dimeric construct of Aβ1–42 (henceforth called $T_{22}$Aβ1–42 flies) and flies expressing the Aβ peptide with the arctic mutation E22G (henceforth called Arctic flies) in the digestive tract of *Drosophila melanogaster*. This *Drosophila* model of AD has previously been developed in our laboratory [40], where the Myo31DF driver was used to specifically express the target protein in the enterocytes of the fly gut.

Our present study revealed that the amount of aggregates increased in both Aβ expressing genotypes as the flies aged, where the expression of $T_{22}$Aβ1–42 resulted in a heavy Aβ load, while the load of aggregates was less prominent in aged Arctic

of authors who received each award: Greta Elovsson Full names of commercial companies that funded the study or authors: This is not a commercial company URLs to sponsors' websites: alzheimerfonden.se The funders had no role in study design, data collection and analysis, decision to publish, or preparation of the manuscript. Grant source: Torsten Söderbergs Stiftelse Specific grant numbers: M44/19 Initials of authors who received each award: K Peter R Nilsson Full names of commercial companies that funded the study or authors: This is not a commercial company URLs to sponsors' websites: Startsida - Torsten Söderbergs Stiftelse The funders had no role in study design, data collection and analysis, decision to publish, or preparation of the manuscript.

**Competing interests:** The authors have declared that no competing interests exist.

flies. Furthermore, by using the two LCO ligands HS-84 and HS-169 to decipher the aggregate staining patterns, we found that HS-84 and HS-169 label Aβ differently depending on whether the Aβ assemblies originate from Arctic Aβ1–42 or $T_{22}$Aβ1–42. HS-84 labeled Arctic Aβ1–42 aggregates better than HS-169, and surprisingly the inverse was seen in $T_{22}$Aβ1–42 flies. A greater load of Aβ species in the $T_{22}$Aβ1–42 flies compared to the Arctic flies was also confirmed through Meso Scale Discovery (MSD) analyses. Since these two Aβ genotypes have similar median survival times, the toxic effect of the combined amount of aggregates is remarkably higher in the Arctic flies compared to the $T_{22}$Aβ1–42 flies. To gain further insight into the stability of the aggregates formed in these flies, the amount of Aβ species that could resist a certain concentration of Gua-HCl was investigated for each fly genotype. Although the total amount of Aβ1–42 was strikingly higher in the $T_{22}$Aβ1–42 flies compared to the Arctic flies, the levels of Aβ1–42 species found in the lowest (2.5 M) [Gua-HCl] – fraction were rather similar for both Aβ genotypes. If exclusively considering these Aβ assemblies, the toxic effect becomes fairly similar between the two genotypes. Assuming the same toxic mechanism for the Arctic and the $T_{22}$Aβ1–42 flies, these aggregated species could be responsible for the toxicity, leading to an equal lifespan reduction. However, the most accumulated species in the Arctic and the $T_{22}$Aβ1–42 flies are Aβ assemblies residing in the 4 M and 5 M Gua-HCl-fraction, respectively, revealing that the aggregation mechanism between these two genotypes differs. This suggests that, while intermediary protofibrillar Aβ assemblies constitute the toxic species for the Arctic flies, the cause of death in $T_{22}$Aβ1–42 flies could be the tremendous amount of presumably mature fibrillar aggregates built up in the gut of these flies, pointing towards that the mechanism of toxicity is different between the two fly genotype despite sharing an equal reduction in the life span. Indeed, an analysis of gut leakage revealed that aggregates from Arctic Aβ1–42 can cause gut leakage, unlike the $T_{22}$Aβ1–42 aggregates. This finding supports the notion that these two Aβ variants might operate through different toxic mechanisms.

## Results

### Age-related increase of Aβ aggregates in Arctic and $T_{22}$Aβ1–42 flies

To address whether the amount of aggregates in the fly's anterior midgut changes over time, the gut of $T_{22}$Aβ1–42 and Arctic flies were stained when the flies were 4 and 8 days old using the LCO ligands HS-84 or HS-169, which bind cross β-sheet structures. The time points were chosen based on the survival assay performed on Arctic and $T_{22}$Aβ1–42 flies from our previous study (see Fig 1A).Day 8 is fairly close to the median survival time for both genotypes [40], marking the point when the flies begin to die. However, there are still enough flies alive to gather the required number for the experiment.Day 4 was selected since it is between eclosion and day 8.

At day 4, positive aggregate staining was detected with HS-84 and HS-169 in the gut of $T_{22}$Aβ1–42 flies (see Fig 1B). As these flies got older, the Aβ load increased relative to what was seen in younger flies (see HS-84 and HS-169, in Fig 1B). The overall picture of the Aβ assemblies originating from the $T_{22}$Aβ1–42 construct is that they are very heterogeneous resulting in an array of different shapes and sizes.

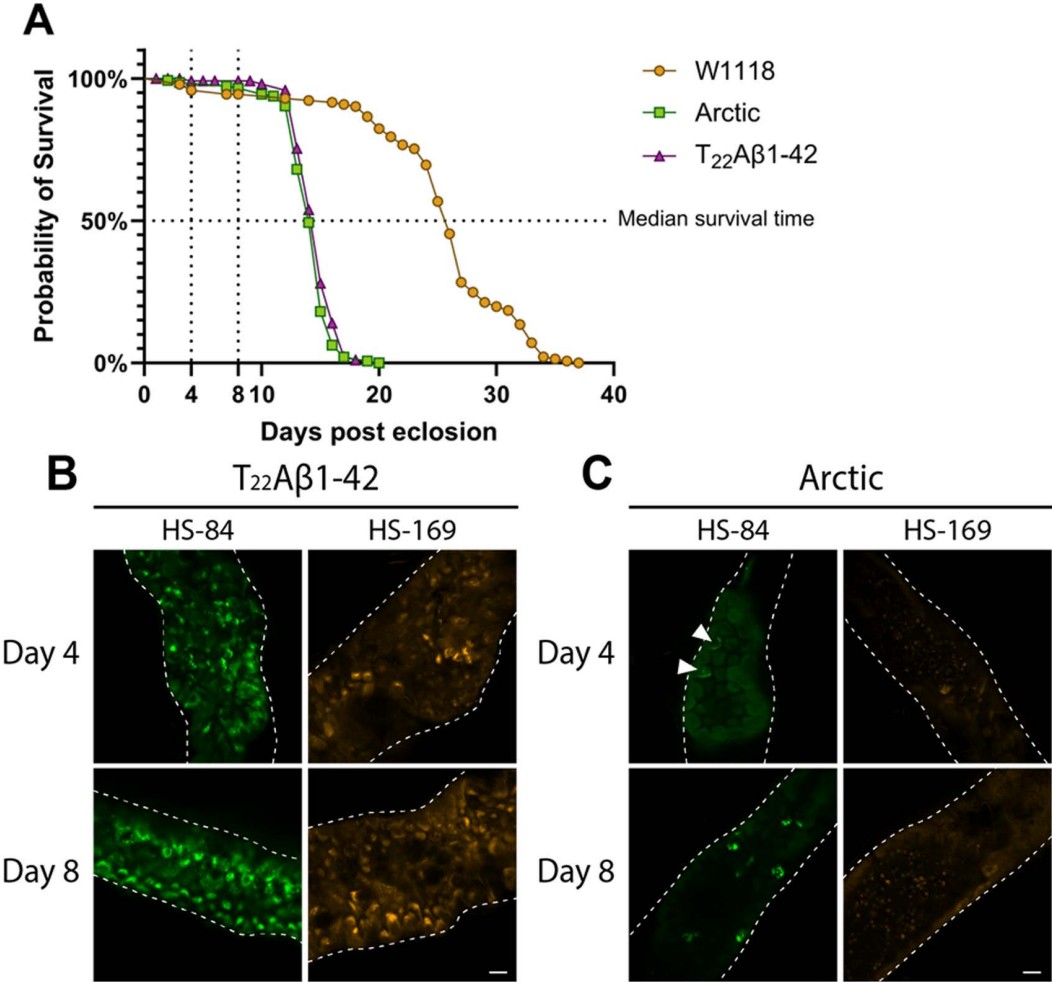

**Fig 1. Age-related increase in the amount of Aβ aggregates in the midgut of Aβ-expressing flies. (A)** Illustration of longevity analyses of the Arctic-, $T_{22}$Aβ1-42- and control w1118 flies, from a previous study [40]. The fluorescence acquisition was carried out when the flies were 4 and 8 days old, see markings. **(B-C)** Age study of flies expressing either (B) the tandem dimeric construct $T_{22}$Aβ1-42 or (C) the Arctic mutant of Aβ1-42 in the enterocytes using the Myo31DF driver line. **(C)** Arrowheads indicate vague outlines of aggregates. The gut tissue is stained with LCOs HS-84 (green) or HS-169 (orange). The age of the flies is indicated in the figure. Scale bars are 20 μm.

Proceeding to the Arctic flies, no clear positive aggregate staining was detected by HS-84 and HS-169 on day 4 (see Fig 1C); only vague outlines of what appears to be Aβ aggregates can be seen with HS-84 (see white arrows). When the flies were 8 days old, HS-84 labeled several arctic Aβ aggregates in the gut, but no distinct staining was seen with HS-169 (see Fig 1C). The age-related increase of positive aggregate signals detected in Arctic flies using HS-84 (compare day 4 with day 8, in Fig 1C) is in line with what was seen in $T_{22}$Aβ1–42 flies, although fewer aggregates were identified in Arctic flies at both time points.

This age study shows that both HS-84 and HS-169 can successfully detect Aβ aggregates in $T_{22}$Aβ1–42 flies, whereas only HS-84 is able to clearly detect Aβ aggregates in Arctic flies. Larger and smaller Aβ assemblies are seen with HS-84 and HS-169 in $T_{22}$Aβ1–42 flies at both time points. Differently sized Aβ assemblies are also found with HS-84 in Arctic flies at day 8, though the total amount of detected aggregates in Arctic flies is substantially lower compared to what was seen in $T_{22}$Aβ1–42 flies. With HS-84, an age-related increase of Aβ species is seen in both Aβ-expressing genotypes.

## Gut leakage was evaluated in Arctic-, T$_{22}$Aβ1–42-, and control flies

To gain a deeper understanding of the toxic events in the two different Aβ-expressing flies, an analysis of intestinal permeability was conducted [41–43]. The flies were fed the blue dye erioglaucine disodium salt. The spreading of the blue dye in the hemocoel and body (known as the smurf phenotype) after ingestion is a sign of gut leakage, see Fig 2A. The result from the smurf assay showed that 3 flies with the smurf phenotype were detected in Arctic flies (n = 38) at day 8, while no smurfs was found at the same timepoint in T$_{22}$Aβ1–42 flies (n = 32) and control flies (n = 44), see Fig 2B.

## LCOs exhibit different staining patterns in Arctic and T$_{22}$Aβ1–42 flies

Next, the heterogeneity among Aβ aggregates produced in the gut of Arctic and T$_{22}$Aβ1–42 flies was studied through co-staining using an antibody against Aβ and the LCO ligands HS-169 and HS-84. To characterize the detected Aβ species, the differences between HS-84 and HS-169 regarding Aβ-labelling were compared. If not specified, the gut region selected for the fluorescence image acquisition was the anterior midgut, as described elsewhere [40].

The aggregates in the gut tissue of T$_{22}$Aβ1–42 flies showed positive Aβ staining with the anti-Aβ antibody, HS-84, and HS-169, albeit to a different extent (see Fig 3). HS-169 and the antibody colocalized almost completely and visualized a great amount of Aβ species in the T$_{22}$Aβ1–42 fly gut. In contrast, HS-84 failed to detect some of the smaller aggregates that were stained by the antibody and HS-169 (see white circles in Fig 3). To ensure that HS-169 did not simply

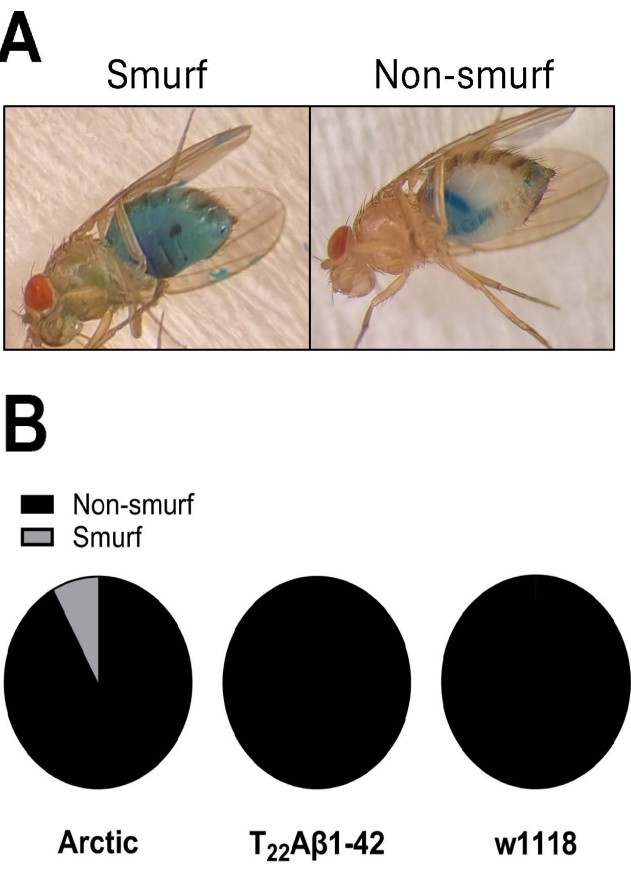

**Fig 2. The smurf phenotype indicates gut leakage. (A)** A fly with the smurf phenotype to the left compared to a non-smurf fly to the right. **(B)** The smurf phenotype was evaluated in Arctic-, T$_{22}$Aβ1-42- and control w1118 flies, 8 days post eclosion.

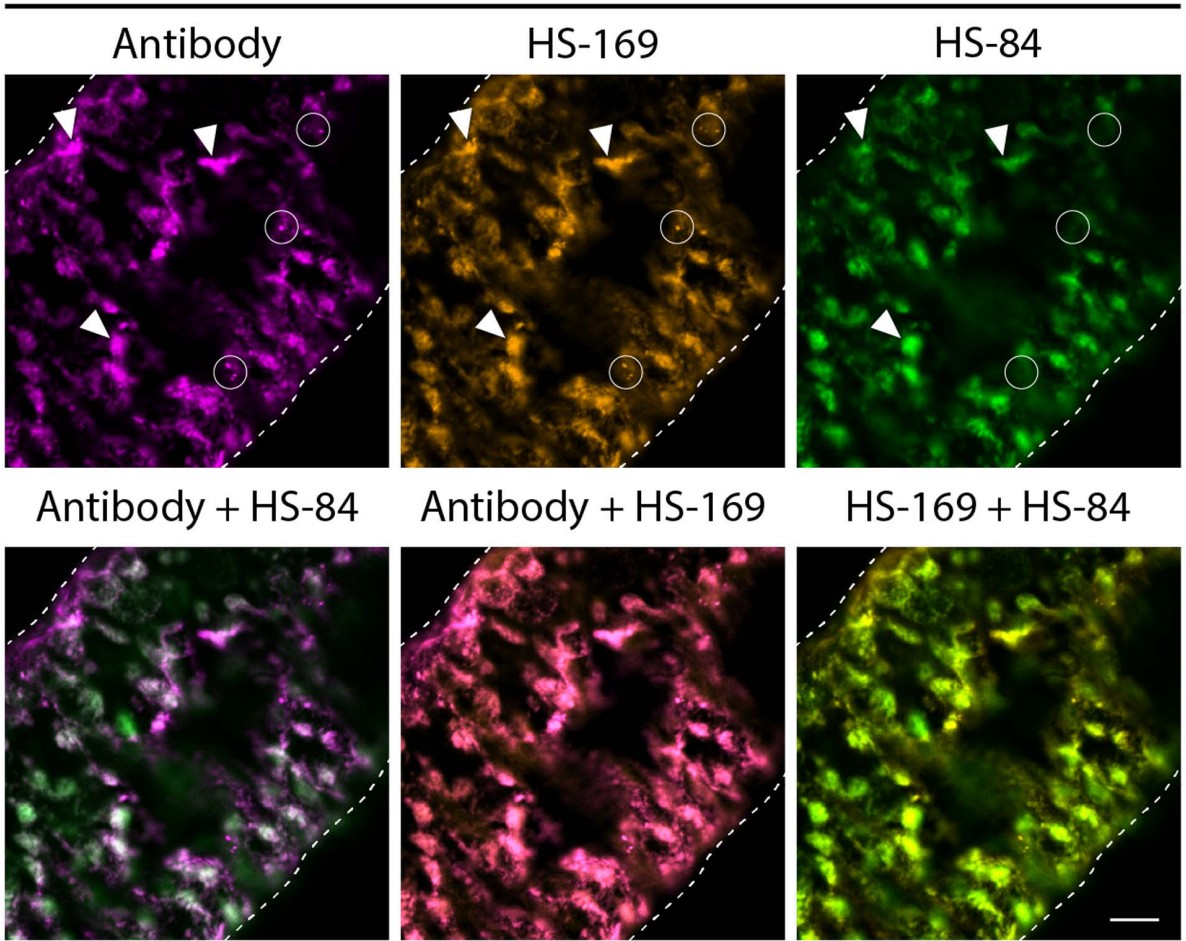

**Fig 3. Several smaller Aβ species detected by the antibody and HS-169 in T22Aβ1-42 flies are not detected by HS-84.** Detection of Aβ species in the anterior midgut of Drosophila flies expressing the tandem construct $T_{22}$Aβ1-42 in the enterocytes using the Myo31DF driver. The confocal microscope single-plane images show gut tissue co-stained with Mabtech anti-human Aβ antibody (magenta) and LCO ligands HS-84 (green) and HS-169 (orange) 8 days post eclosion. The white, dotted line outlines the edges of the intestine. White circles: Aβ species labeled by the antibody and HS-169. White arrowheads: Aβ species labeled by all three staining agents. Scale bar, 20 μm.

outcompete HS-84 for the binding sites on these smaller aggregates a co-staining with only the antibody and HS-84 was executed (see S1 Fig). We found similar Aβ antibody-positive species that were negative with HS-84, proposing that HS-169 did not outcompete HS-84 in this case. Like in the age study, a plethora of different aggregates were visible in $T_{22}$Aβ1–42 flies, and the majority of them – the Aβ bulk – was detected by all three staining agents (see examples, white arrowheads in Fig 3).

Considerably fewer positive Aβ signals were identified in Arctic flies relative to $T_{22}$Aβ1–42 flies (see Fig 4). Indeed, quantifications of the co-stainings showed that the number of Aβ species per mm$^2$ was approximately 40x higher in $T_{22}$Aβ1–42 flies compared to Arctic flies, see Fig 5. As was found in the age study, HS-169 had limited abilities to stain aggregates in Arctic flies, while HS-84 and the antibody colocalized almost completely, confirming that HS-84 labeled Arctic Aβ1–42 considerably better than HS-169 did (see white circles in Fig 4). There were aggregates in the Arctic flies that

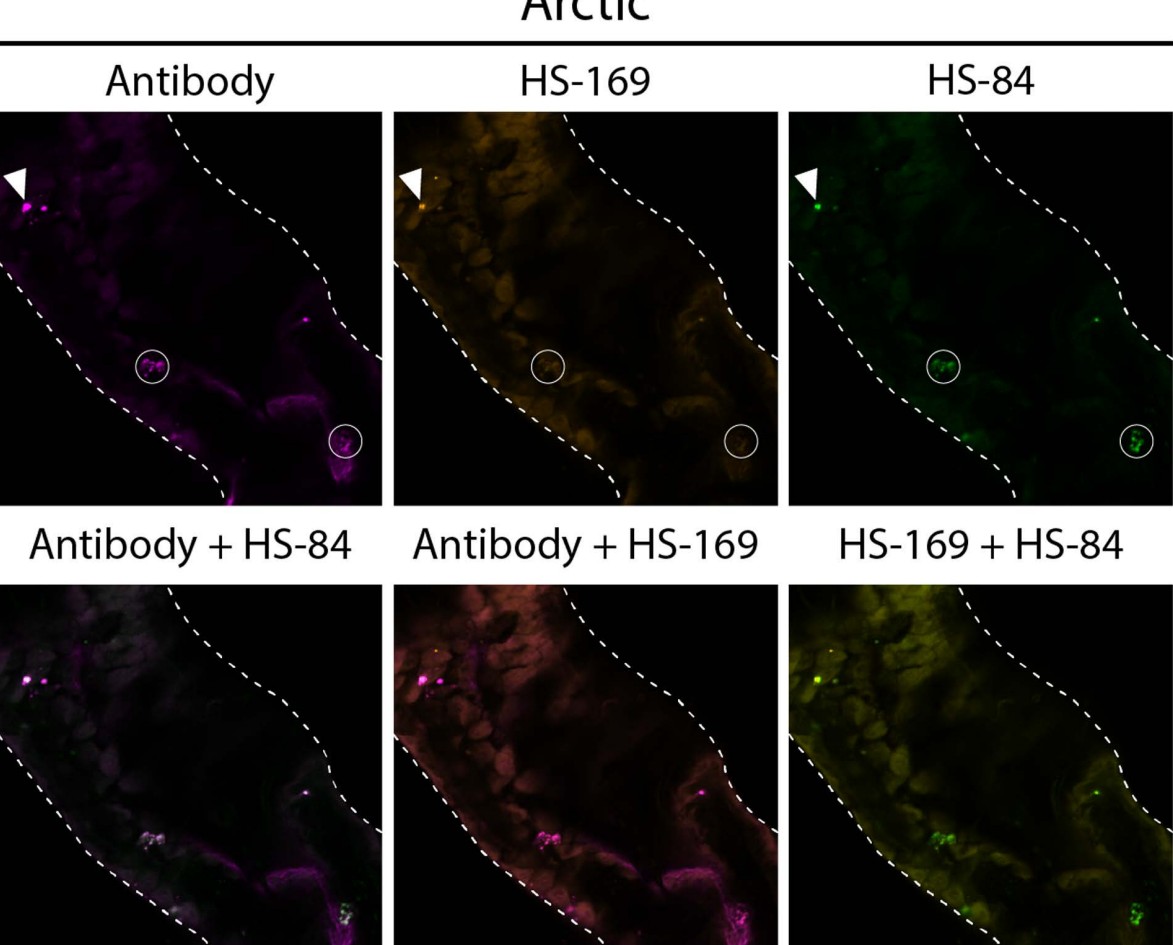

**Fig 4. Several Aβ species detected by the antibody and HS-84 in Arctic flies are not detected by HS-169.** Detection of Aβ species in the anterior midgut of Drosophila flies expressing the Arctic mutant of Aβ1-42 in the enterocytes using the Myo31DF driver. The confocal microscope single-plane images show gut tissue costained with Mabtech anti-human Aβ antibody (magenta) and LCO ligands HS-84 (green) and HS-169 (orange) 8 days post eclosion. The white, dotted line outlines the edges of the intestine. White circles: Aβ species labeled by the antibody and HS-84. White arrowheads: Aβ species labeled by all three staining agents. Scale bar, 20 μm.

the antibody, HS-84, and HS-169 bound to (see white arrowheads in Fig 4 and S2 Fig). However, the bulk of aggregates detected by all three staining agents in $T_{22}$Aβ1–42 flies were not seen in Arctic flies.

In summary, the co-staining results reveal that $T_{22}$Aβ1–42 flies exhibit a 40x higher load of Aβ aggregates compared to Arctic flies. Additionally, HS-169 had more positive aggregate staining compared to HS-84 in $T_{22}$Aβ1–42 flies, whereas the opposite was seen in Arctic flies. To conclude, the aggregates formed in these two Aβ genotypes are stained differently with the LCO ligands.

### The proteotoxic effect of Aβ1–42 is substantially stronger in Arctic flies compared to $T_{22}$Aβ1–42 flies

Since there is a close relationship between Aβ deposition and AD, the MSD technique was used to analyze Aβ1–42 levels in the body of Arctic and $T_{22}$Aβ1–42 flies. The experiment was conducted when the flies were 8 days old, which

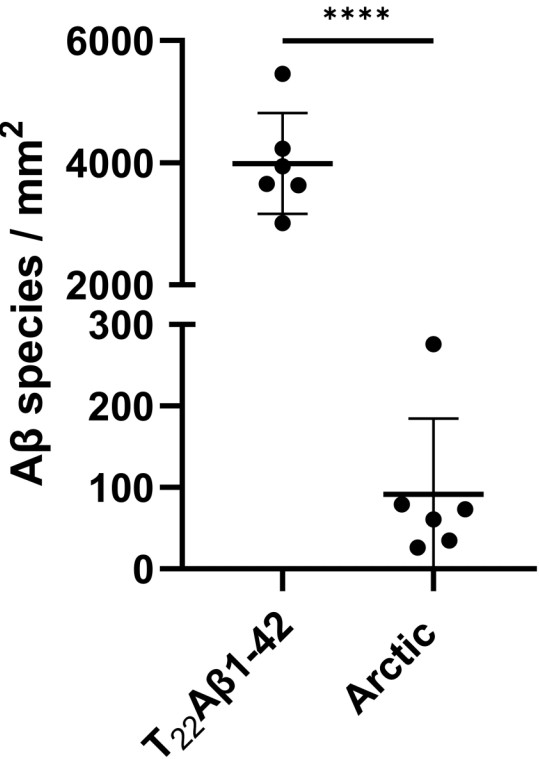

**Fig 5. Quantifications of the co-staining in T22Aβ1-42- and Arctic flies.** Nonbiased scoring of Aβ1-42 species per mm² in Arctic- and T$_{22}$Aβ1-42 flies at day 8, n=6. Only Aβ1-42 species where HS-84 or HS-169 colocalized with the antibody were counted. Data represented as ± SD. **** represents P<0.0001, respectively, as determined by an unpaired t test. ns equals non-significant.

corresponded to a time point close to the median survival time for the two genotypes. First, Aβ species soluble in buffer containing 0 M Gua-HCl were extracted from the fly sample (=> soluble Aβ1–42). Next, the fly sample was treated with 5 M Gua-HCl to extract those aggregates that were not soluble in 0 M Gua-HCl (=> insoluble Aβ1–42).

The study showed that the T$_{22}$Aβ1–42 flies had the greatest amount of soluble Aβ1–42 (0.70±0.02 pg/fly), whereas the Arctic flies only contained negligible amounts (0.01±0 pg/fly) (see Fig 6A). The levels of insoluble Aβ1–42 followed a similar trend where the highest amount of insoluble Aβ1–42 was found in the T$_{22}$Aβ1–42 flies (10.64±2.48 pg/fly), while the level was lower in the Arctic flies (2.68±0.08 pg/fly) (see Fig 6B). Insignificant levels of Aβ1–42 were detected in the W1118 control flies (soluble Aβ1–42: 0±0 pg/fly; insoluble Aβ1–42: 0.01±0 pg/fly). The total Aβ1–42 level (soluble+insoluble Aβ1–42) was calculated to be 2.69 pg/fly in Arctic flies and 11.34 pg/fly in T$_{22}$Aβ1–42 flies. These data are consistent with the results from the quantification of the co-staining, which revealed a higher Aβ load in T22Aβ1–42 flies compared to Arctic flies. Interestingly, despite acquiring a similar toxic effect in terms of the reduction in the median survival time (11 days for T$_{22}$Aβ1–42 flies and 12 days for Arctic flies [40]) the total amount of Aβ1–42 species is much higher in the T$_{22}$Aβ1–42 flies compared to the Arctic flies. The proteotoxic effect was assessed from these data by dividing the toxic effect – reduction in median survival time relative to W1118 control flies – with the total amount of Aβ1–42 per fly. Here we found that the proteotoxic effect in Arctic flies (4.5 days/pg Aβ1–42) was more than 4 times higher than for T$_{22}$Aβ1–42 flies (0.97 days/pg Aβ1–42), see Fig 6C.

In summary, the MSD experiment revealed that: (1) both Aβ-expressing genotypes have higher levels of insoluble Aβ1–42 than soluble Aβ1–42; (2) T$_{22}$Aβ1–42 flies have a higher level of total Aβ1–42 compared to what was detected in

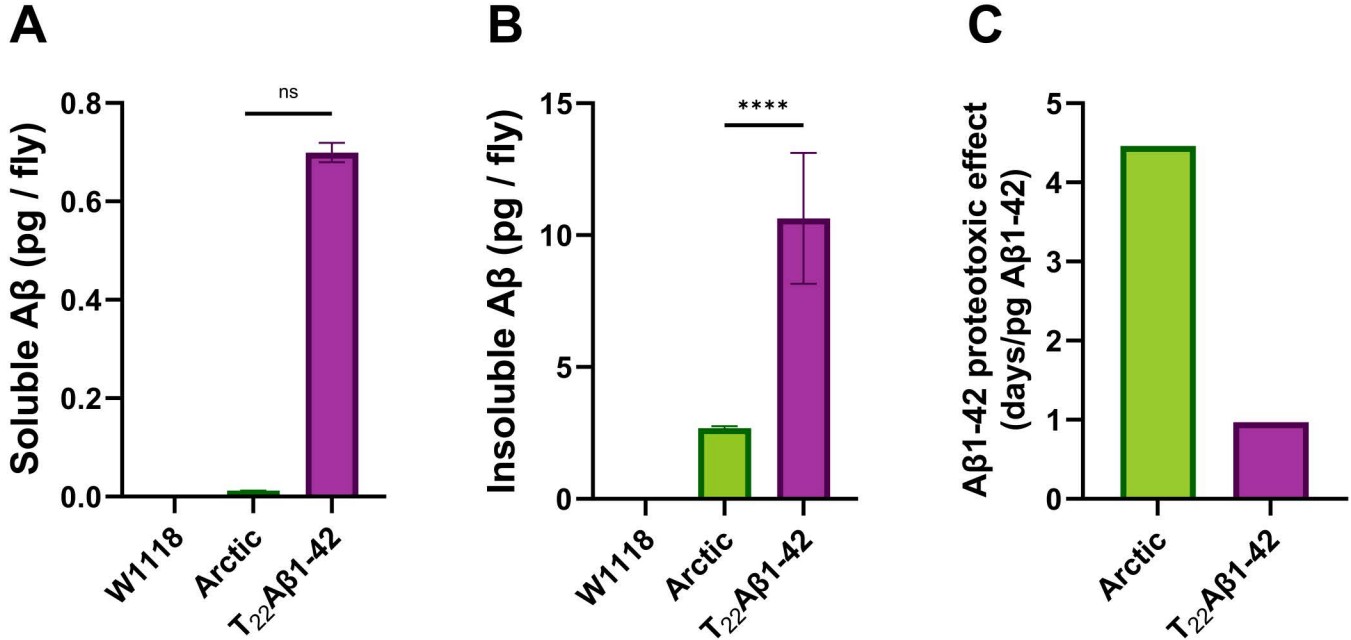

**Fig 6. The proteotoxic effect is substantially stronger in Arctic flies compared to T22Aβ1-42 flies.** MSD analyses were performed on the body of Myo31DF-derived flies expressing the Arctic mutant of Aβ1-42 or tandem construct $T_{22}$Aβ1-42 to measure the levels of Aβ1-42. Data represented as mean (±s.d). **(A)** The levels of soluble Aβ1-42. **(B)** The levels of insoluble Aβ1-42. **(C)** The Aβ1-42 proteotoxic effect was assessed by dividing the reduction in median survival time of the Aβ genotypes relative to the control flies, with the total level of Aβ1-42 per fly (see A-B). Data represented as ± SD. **** represents P < 0.0001 as determined by a one-way ANOVA followed by Tukey's test. ns equals non-significant.

Arctic flies; (3) the proteotoxic effect is significantly higher for the Aβ species formed in Arctic flies than those formed in $T_{22}$Aβ1–42 flies.

### Gua-HCl gradient reveals different stability among aggregates produced in Arctic and $T_{22}$Aβ1–42 flies

To gather more information about the aggregates formed in Arctic and $T_{22}$Aβ1–42 flies and to identify properties associated with toxicity, the insoluble Aβ1–42 fraction was further divided by gradually extracting the aggregates from the fly body with increasing concentrations of Gua-HCl: 0 M, 2.5 M, 4 M, and 5 M (the fractions are named accordingly). Using a Gua-HCl gradient enabled us to differentiate between aggregates with different stability against the denaturant.

Data from this analysis showed that Aβ1–42 species from $T_{22}$Aβ1–42 flies were detected in all four fractions (0 M, 2.5 M, 4 M, and 5 M), while Aβ species in the Arctic flies could only be found in fractions treated with Gua-HCl (see Fig 7A and Table 1). Interestingly, it was found that the 2.5 M Gua-HCl fraction contained similar levels of Aβ1–42 in both Arctic and $T_{22}$Aβ1–42 flies. To further evaluate these MSD results, a percentage amount was calculated by dividing the amount of Aβ1–42 in each Gua-HCl fraction by the total amount of Aβ1–42 for each fly genotype (see Fig 7B). In $T_{22}$Aβ1–42 flies, the Aβ1–42 levels decreased with descending concentrations of Gua-HCl: 5 M (59%), 4 M (30%), 2.5 M (8.2%), 0 M (2.2%). The Arctic flies, on the other hand, had the most Aβ1–42 in the 4 M fraction (48%), followed by 5 M (36%), 2.5 M (17%), and lastly 0 M (0.28%) representing an insignificant amount. The percentage of Aβ1–42 that populates 2.5 M or 4 M-fraction is considerably larger in Arctic flies compared to $T_{22}$Aβ1–42 flies. Lastly, the difference in Aβ1–42 levels between Arctic- and $T_{22}$Aβ1–42 flies was determined for each fraction (see Fig 7C), which was found to be smaller for the 2.5 M and 4 M fractions compared to the 5 M fractions.

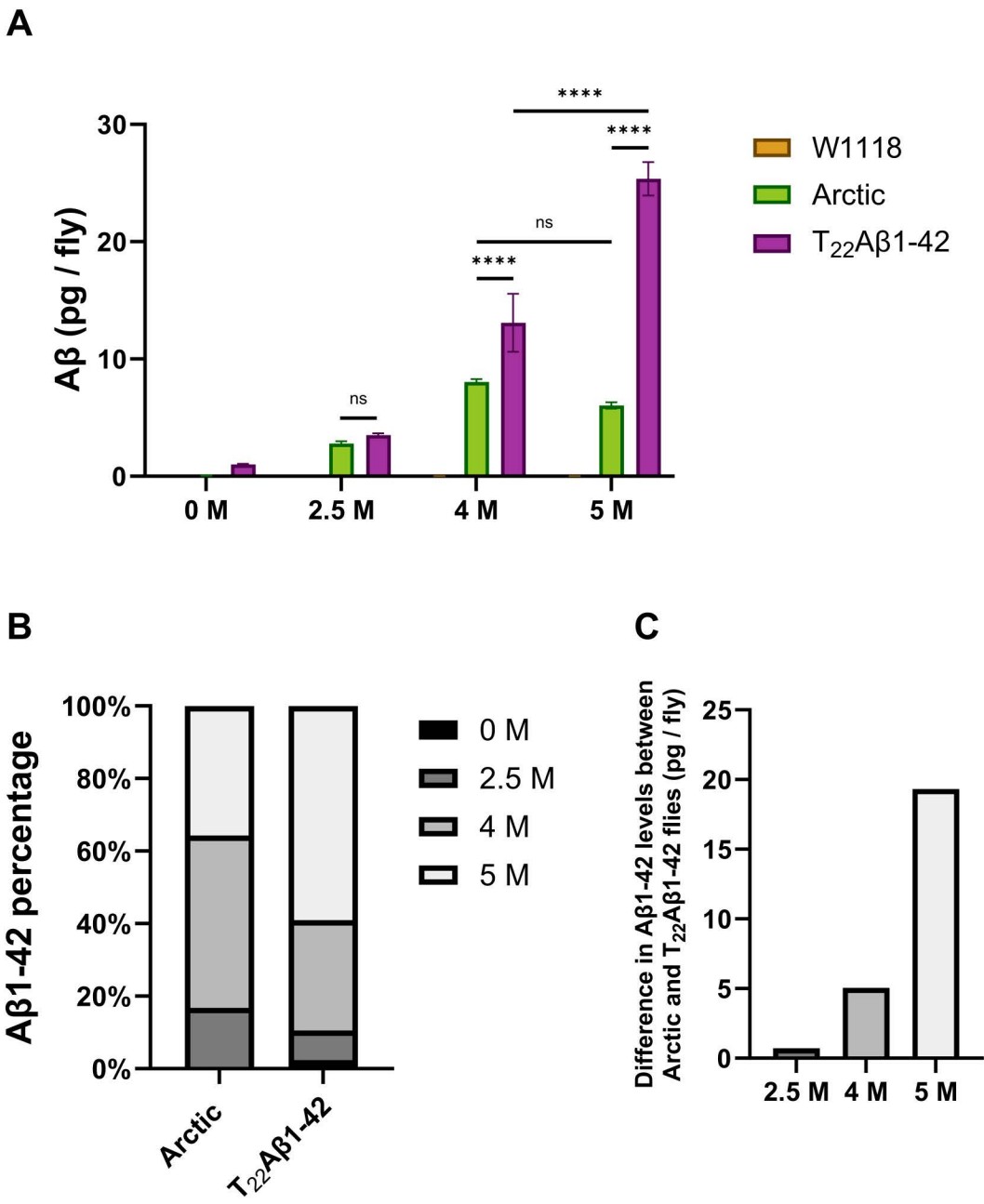

**Fig 7. MSD analyses show different Aβ species with varying stability.** MSD analyses were performed on the body of Myo31DF-derived flies expressing the Arctic mutant of Aβ1-42 or tandem construct T$_{22}$Aβ1-42 to measure the levels of Aβ1-42. Data represented as mean (±s.d). **(A)** Samples were treated with increasing concentrations of Gua-HCl: 0 M; 2.5 M; 4 M; and 5 M, to obtain the level of Aβ1-42 in each fraction (named according to the Gua-HCl concentration). Data represented as ± SD. **** represents $P < 0.0001$ as determined by a one-way ANOVA followed by Tukey's test. ns equals non-significant. **(B)** The amount of Aβ1-42 is calculated in percentage for each fraction relative to the total level of Aβ1-42. **(C)** The difference in Aβ1-42 levels between Arctic and T$_{22}$Aβ1-42 flies is calculated for all Gua-HCl fractions (0 M-, 2.5 M-, 4 M-, and 5 **M)** (data is seen in Fig 7A and Table 1).

**Table 1. Statistic data from MSD analysis where the stability of Aβ aggregates produced in Arctic and T$_{22}$Aβ1-42 was tested against Gua-HCl.**
SD: standard deviation. Aβ1-42 levels in the 0 M-, 2.5 M-, 4 M-, and 5 M fractions were measured (graph illustrated in Fig 7A).

| Gua-HCl conc. (M) | W1118 | | Arctic | | T$_{22}$Aβ1–42 | |
|---|---|---|---|---|---|---|
| | Mean (pg/fly) | SD | Mean (pg/fly) | SD | Mean (pg/fly) | SD |
| 0 | 0.00 | 0.01 | 0.05 | 0.01 | 1.02 | 0.04 |
| 2.5 | 0.02 | 0.01 | 2.79 | 0.20 | 3.52 | 0.13 |
| 4 | 0.01 | 0.01 | 8.04 | 0.24 | 13.09 | 2.48 |
| 5 | 0.03 | 0.01 | 6.04 | 0.28 | 25.36 | 1.42 |

## Discussion

According to the current consensus, amyloid deposits are built up in an AD brain over time [15,41,42]. Amyloid plaques with different morphologies can be found in postmortem human brain tissue, where some are dense with a clear core, while others are more diffuse [43–45]. Since Aβ is the main component in amyloid plaques, studying its aggregation behavior, structures, and toxic properties will advance our understanding of AD. Overproduction of Aβ typically leads to a higher level of toxicity [27,28,46]; however, this is not always the case. Several *Drosophila* studies show that Aβ aggregates can be prominent in the fly brain and still not cause harm [29,47]. This suggests that instead of only focusing on the entire Aβ load as the toxic entity, one must identify the toxic Aβ species and target them to provide a well-thought-out therapeutic approach. Investigating the properties of Aβ aggregates in an *in vivo* system is of great importance to gain further insights into the toxic mechanisms behind Alzheimer's disease. The Aβ aggregation process is very complex since it generates different kinds of species. Initially, the Aβ fibrils were believed to be the toxic contributors to AD, which was later determined not to be the case. Instead, evidence suggests that the oligomers and protofibrils are the pathological drivers that greatly participate in neurotoxic mechanisms [14–17,18]. In this study, variations within the built-up pool of Aβ aggregates were monitored using a *Drosophila* model of AD, in which the Arctic mutant of Aβ1–42 and the tandem construct T$_{22}$Aβ1–42 were expressed in the fly gut. First, an age study was performed where the gut of Arctic and T$_{22}$Aβ1–42 flies was stained with LCOs HS-84 and HS-169, at day 4 and 8. The first observation was that T$_{22}$Aβ1–42 flies had a significantly greater Aβ load compared to the Arctic flies, at both time points, confirming results from our previous paper [40]. We also found that the amount of aggregates increased in both Aβ genotypes as the flies aged, demonstrating a build-up of aggregates in the fly gut, resulting in the death of the flies. In line with the observation in our AD fly model, an age-related/dependent increase in amyloid deposition has also been seen in other AD studies [27,41,48,49]. The array of different shapes and sizes among the Aβ aggregates detected in the gut of Arctic and T$_{22}$Aβ1–42 flies reveals a heterogeneity in the formed Aβ species that can also be observed in human AD brain tissue [43–45], thus validating *Drosophila melanogaster* as a suitable model organism for studying Aβ aggregate heterogeneity related to AD.

To further analyze the diversity of Aβ aggregates formed in Arctic and T$_{22}$Aβ1–42 flies, the gut was co-stained with an anti-Aβ antibody, HS-84, and HS-169. Note that the antibody recognizes the Aβ epitope sequence 3–8; meanwhile, HS-84 and HS-169 bind to the cross-β-sheet structure, which is defined as the repeated pattern of β-sheets found in amyloid fibrils. The binding of HS-84 and HS-169 depends, for example, on the architecture of the cross-β-sheet structures. The antibody staining pattern revealed an Aβ bulk in the T$_{22}$Aβ1–42 flies similar to what was found by HS-84 in the age study, confirming the build-up of Aβ assemblies in these flies. As expected, these Aβ species were also detected by HS-84 and HS-169 in the co-staining experiment. Interestingly, antibody staining of T$_{22}$Aβ1–42 flies also uncovered smaller aggregates, which were identified by only HS-169 and not HS-84. Since HS-169 and HS-84 partially compete for the same binding site, we needed to ensure that this discovery was not because HS-169 outcompeted HS-84. Therefore T$_{22}$Aβ1–42 fly guts were co-stained with only HS-84 and an anti-Aβ antibody. Data from this experiment exhibited similar accumulations of smaller Aβ species that were labeled by the antibody but not HS-84, confirming that HS-84 is not outcompeted by

HS-169 for the binding site of these smaller aggregates. Both HS-169 and HS-84 bind cross-β-sheet structures, however, the fact that these two amyloid ligands do not completely co-localize suggests that these smaller aggregates are different from the Aβ bulk that were labeled by all three staining agents in the $T_{22}$Aβ1–42 flies. When studying the stained gut of Arctic flies, the antibody and HS-84 could only identify a small amount of aggregates, which were not stained by HS-169. The large bulk of Aβ assemblies that were visualized in $T_{22}$Aβ1–42 flies was not seen in Arctic flies. In Arctic flies, HS-84 was able to detect several aggregates, while HS-169 had a limited staining pattern which was in line with the results from the age study. Thus, it appears that HS-84 and HS-169 have different staining patterns in Arctic and $T_{22}$Aβ1–42 flies, revealing that the pools of the aggregates formed in these flies are different. First, we have a large bulk of Aβ aggregates in the $T_{22}$Aβ1–42 flies that could be identified by both LCOs and was absent in the Arctic flies. However, in each Arctic fly gut, there were a few areas with positive signals from all three staining agents. Then, we have the presence of other types of Aβ species that were solely HS-169 positive in the $T_{22}$Aβ1–42 flies or HS-84 positive in the Arctic flies. That HS-84 and HS-169 only partially co-localize and thus can distinguish different kinds of aggregates has been reported in a previous study, where they evaluated *in vivo* and *ex vivo* staining patterns of HS-84 and HS-169 in mice [34]. They found that HS-169 labeled the plaque core more vividly, while HS-84 showed a more diffuse aggregate binding pattern. These results, in addition to ours, point towards the fact that different types of Aβ deposits can be identified by HS-84 and HS-169. This diverse labeling of amyloid dyes, as a consequence of aggregate heterogeneity, has been seen in other studies as well [32,33]. It is known that different packing of prefibrillar species and mature Aβ fibrils affects the binding of molecular probes [32,50]. An amyloid ligand with a rigid construction must fit quite perfectly on the binding surface or the interaction will fail and the target will not be detected. Due to structural differences, HS-169 is less flexible than HS-84 and thus has more restrictions when it comes to labeling targets [38]. The fact that the aggregate binding pattern of HS-84 and HS-169 differs between the $T_{22}$Aβ1–42- and Arctic flies indicates that the Aβ aggregates formed by the $T_{22}$Aβ1–42 construct and the Arctic Aβ1–42 peptide *in vivo* might be different. One can speculate whether the aggregates with positive Aβ staining from all three agents in Arctic and $T_{22}$Aβ1–42 flies are similar. Our conclusion is, though, that the aggregates in question have different appearances and that they, therefore, likely do not maintain the same properties.

To quantitatively determine the level of soluble and insoluble Aβ1–42 in Arctic and $T_{22}$Aβ1–42 flies, MSD analyses were performed when the flies were 8 days old. Here, we observed that the total level of Aβ1–42 was substantially lower in flies expressing the Arctic peptide compared to flies expressing the tandem peptide, which agreed well with the results from the optical images. When studying the Aβ1–42 levels further, we found that insoluble Aβ1–42 was more abundant than soluble Aβ1–42 in both fly genotypes, revealing that most of the Aβ peptide produced in these flies have aggregated. In fact, the soluble level in the Arctic flies was not even within the detection limit of the measurement. To compare toxic effects between different assemblies of Aβ aggregates, the proteotoxic effect can be calculated where the reduction in median survival time of a fly genotype (relative to its control) is divided by the entire Aβ load per fly. Although the survival data revealed a similar reduction in median lifespan between the two fly genotypes, a potential saturation effect of Aβ accumulation should be considered which could mean that once Aβ levels reach a certain threshold, additional accumulation does not lead to increased toxicity. However, the outcome from the calculation of the proteotoxic effect showed that the Aβ1–42 pool produced in Arctic flies was 4 times more proteotoxic than the Aβ1–42 pool produced in $T_{22}$Aβ1–42 flies revealing that the toxic effect per Aβ unit (pg/fly) is higher in the Arctic flies compared to the $T_{22}$Aβ1–42 flies. These results are consistent with our previous paper, where we demonstrated that the amount of Aβ aggregates did not correlate with the level of toxicity (mortality) in Arctic and $T_{22}$Aβ1–42 flies, and other studies have made similar observations. Luheshi and her team evaluated the toxic effects of the Arctic (E22G) Aβ1–42 peptide – with and without the I31E mutation [47]. They found that both Arctic Aβ variants formed aggregates at similar rates, however, the I31E mutation extended the lifespan of flies expressing Arctic Aβ1–42 despite an extensive Aβ deposition. Similar findings were made when a tandem Aβ1–40 construct (two Aβ1–40 peptides conjoined by a linker of 12 amino acids) was expressed in the fly brain resulting in the accumulation of aggregates without any toxic effects in the longevity assay [29]. Instead, different Aβ species likely

exert various levels of toxicity. To gain a more detailed view of the composition of Aβ species in the insoluble pool of Aβ, aggregates in the flies were gradually dissolved using an ascending concentration of Gua-HCl: 0 M, 2.5 M, 4 M, and 5 M. Aβ species residing in fractions with higher Gua-HCl concentrations are arguably more stable than those dissolved in a lower molarity of Gua-HCl. The data showed that Aβ1–42 was present in all four fractions in $T_{22}$Aβ1–42 flies, while Aβ species originating from the Arctic Aβ1–42 were only found in fractions treated with Gua-HCl. Assuming that the Aβ species residing in each fraction are structurally distinguishable, we can state that the $T_{22}$Aβ1–42 flies have at least four different kinds of Aβ assemblies while the Arctic flies have three. This difference suggests a slightly wider variance of aggregates, with different stability, in $T_{22}$Aβ1–42 flies than in Arctic flies. It is established that the MSD analysis, using a Gua-HCl gradient, is capable of detecting up to four different structures, whereas the immunohistochemical assay detects two kinds of aggregates. One could assume that the MSD fraction with the most Aβ is populated by the most prominent aggregate type in the immunohistochemical assay. This implies that the tandem Aβ species exerted in 5 M Gua-HCl are attributed to the large bulk of insoluble Aβ aggregates detected in $T_{22}$Aβ1–42 flies, while the Arctic Aβ species in the 4 M fraction belong to the aggregates detected by HS-84 and the antibody in the Arctic flies.

The Aβ1–42 levels in $T_{22}$Aβ1–42 flies increased with ascending Gua-HCl concentrations, resulting in most Aβ1–42 in the 5 M fraction. In contrast, the Arctic flies instead had most Aβ1–42 in the 4 M fraction. A percentage was determined for each fraction where the Aβ1–42 level in that specific fraction was compared against the total amount for that genotype. The percentage amount of Aβ1–42 that populated 2.5 M and 4 M-fraction was considerably larger in Arctic flies (65%) compared to $T_{22}$Aβ1–42 flies (38%), indicating that the larger part of insoluble Aβ species formed in Arctic flies break up more easily than the largest part does in the $T_{22}$Aβ1–42 flies. This result reveals that the two Aβ peptides have different aggregation behavior *in vivo*. While the equilibrium in the aggregation process of the Arctic peptide is shifted towards the 4 M fraction, the aggregation process of the $T_{22}$Aβ1–42 peptide is shifted toward the 5 M fraction. Interestingly, when comparing the actual level of Aβ between Arctic and $T_{22}$Aβ1–42 flies for the various Gua-HCl fractions, the difference is much more prominent in the 5 M fraction compared to the 2.5 and 4 M fraction. The amount of Aβ1–42 is basically equal in the 2.5 M fraction for both Aβ genotypes. The Aβ species found in the 2.5 and 4 M fractions are surely less stable than those dissolved at 5 M Gua-HCl, but yet stable enough not to break down and integrate the soluble 0 M fraction. The most stable and insoluble form included in the Aβ aggregation process is the Aβ fibril. Therefore, the Aβ species in the 5 M fraction are presumably Aβ fibrils, whereas the 2.5 M and 4 M fractions likely consist of prefibrillar species. Thus, if one were to assume that the toxic mechanism is the same in Arctic and $T_{22}$Aβ1–42 flies, then it would suggest that it is those prefibrillar species, dissolving in 2.5 M Gua-HCl, that are responsible for the toxicity in Arctic and the $T_{22}$Aβ1–42 flies leading to a similar reduction in the life span. One can speculate that the Aβ assemblies belonging to the 2.5 M or the 4 M fraction could be products of secondary nucleation. Secondary nucleation is described as the catalytic transformation of free Aβ monomers to nuclei on an already existing amyloid fibril, thus promoting oligomerization, see Fig 8 [19–21]. This catalytic cycle can be inhibited by occupying the binding surfaces on the fibril, e.g., with chaperone domain BRICHOS, and thereby shifting the aggregation process away from toxic intermediates [20,51,52]. Primary nucleation dominates in the beginning when free monomers are common and fibrillar species are scarce, but as the aggregation progresses and builds up a surplus of amyloid fibrils, secondary nucleation becomes more prevalent. Prefibrillar species generated from secondary nucleation could either be involved in the on-pathway growth of Aβ assemblies into fibrils, or they could be used to create off-pathway species. Relating this knowledge to our study, we can declare that Arctic and $T_{22}$Aβ1–42 species from the 2.5 M fraction could be formed on-pathway through both primary- and secondary nucleation, or they might be formed off-pathway through secondary nucleation. Both the Arctic Aβ1–42 peptide and the dimeric construct $T_{22}$Aβ1–42 are continuously expressed in the gut throughout the fly's lifetime, meaning that monomers that participate in primary- and secondary nucleation are frequently added to the aggregation machinery. The products of secondary nucleation could have different structures compared to their primary-nucleated counterparts [20].

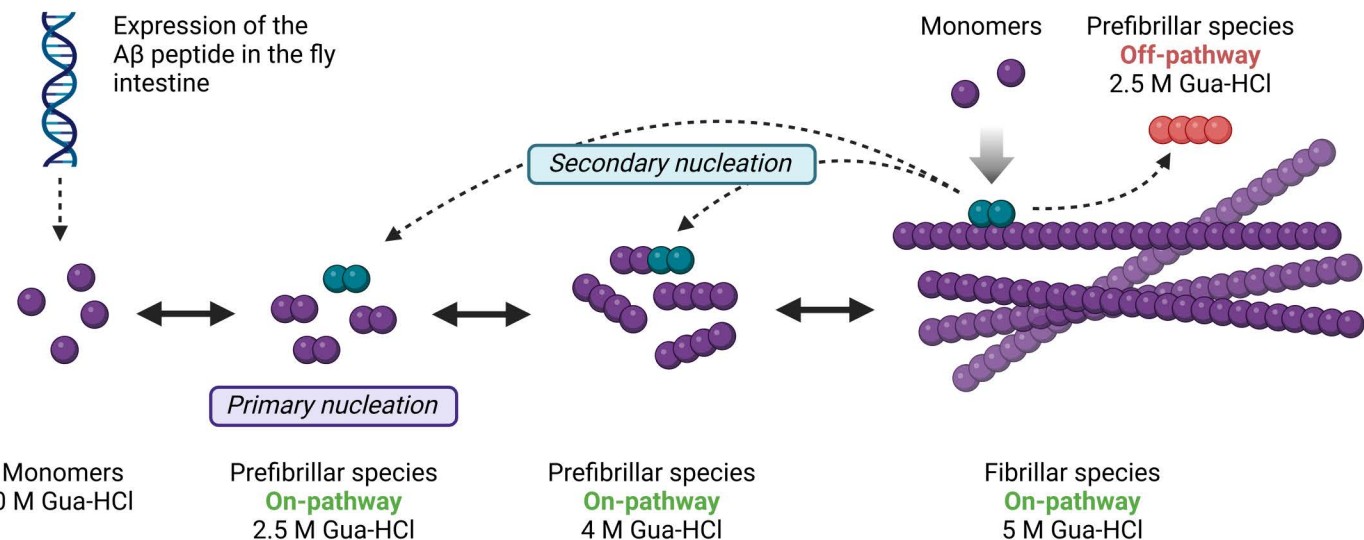

**Fig 8. Proposed illustration of the Aβ fibril formation in relation to the MSD analysis using a Gua-HCl gradient.** Aβ can assemble through primary- or secondary nucleation. The type of Aβ species residing in each Gua-HCl fraction is indicated in the figure.

If the toxicity operates instead through different mechanisms in Arctic and $T_{22}$Aβ1–42 flies, despite having a similar lifespan, the fraction with the most Aβ likely contains the toxic species (4 M for the Arctic flies and 5 M for the $T_{22}$Aβ1–42 flies). Indeed, the smurf assay indicates that Arctic flies are prone to gut leakage, a phenomenon absent in the $T_{22}$Aβ1–42 flies, and thus supporting the notion that these two Aβ genotypes might operate through different toxic mechanisms.

This means that the mechanism of toxicity in the $T_{22}$Aβ1–42 flies could be due to the massive load of aggregates found in these flies, accompanied by a collapse of the intestine. Toxic effects originating from an overload of aggregates have been seen in lysozyme amyloidosis [53–55]. The toxic species in the Arctic flies mightbe the fraction at 4 M since this fraction dominates the formed Aβ structures and could potentially be the cause of the gut leakage. Like the prefibrillar species at 2.5 M, the 4 M structures can be formed through on-pathway primary nucleation or on/off-pathway secondary nucleation. The hypothesis that the prefibrillar species are the main contributors to toxicity is consistent with previous research where oligomers and protofibrils are announced as drivers of AD pathology and progression [14–17,18].

To conclude, expression of the Arctic mutant Aβ1–42 and the dimeric construct $T_{22}$Aβ1–42 resulted in an array of various Aβ species with different ligand binding characteristics and biochemical properties, which were distinguished by optical imaging using antibodies and LCOs and through MSD analyses. Assuming that the toxic mechanisms in Arctic and $T_{22}$Aβ1–42 flies differ, the cause of death could be due to the most prominent Aβ structures found in these flies; a massive load of insoluble aggregates in $T_{22}$Aβ1–42 flies, and prefibrillar species in Arctic flies. However, it cannot be excluded that the toxicity of both Arctic and $T_{22}$Aβ1–42 flies might be attributed to a group of easily dissolved Aβ species, pointing towards a similar pathological process in the two fly genotypes where intermediate species are responsible for the death of the flies. Either way, this study provides valuable information that two different Aβ peptides, resulting in equal toxic effects when expressed in the intestine of the flies, display an array of different Aβ assemblies, revealing the plethora of various aggregated species that the Aβ peptide is able to convert into.

## Materials and methods

### Drosophila stocks

The Gal4/UAS was exploited to accomplish cell-specific expression in the enterocytes using the Myo31DF fly driver line. Fly lines carrying Aβ were kindly provided by D. Crowther (AstraZeneca, Floceleris, Oxbridge Solutions Ltd., London, United Kingdom). These fly lines include the arctic mutant of the Aβ1–42 peptide (Arctic flies) and the tandem construct $T_{22}$Aβ1–42 ($T_{22}$Aβ1–42 flies) in which two Aβ1–42 peptides are joined by a 22 amino acid long linker. The expressed Aβ peptide or construct has a secretory signal peptide [27,29]. Control w1118 flies were purchased at Bloomington Stock Center. Fly crosses and offspring were maintained at 25° and 29°, respectively, at 60% humidity with 12:12-h light: dark cycles. Female flies were selected and were transferred into new vials with fresh fly food every 2–3 days. The offspring were dissected 4 and 8 days post eclosion for the age study, and 8 days post eclosion for the costaining. The offspring were snap-frozen 8 days post eclosion for the MSD analyses.

### Smurf assay

When the flies were 7 days old they were moved to instant fly food (Scientific Laboratory Supplies) containing 2.5% erioglaucine disodium salt and kept there overnight [41–43]. The smurf phenotype of the flies was analysed the next day, where spreading of the blue dye in the hemocoel and body indicated gut leakage. The gut leakage analysis was performed when the flies were 8 days old.

### Age study

Dissected intestines were fixed in 4% formaldehyde (Merck, Darmstadt, Germany) for 10 min at RT, washed in PBS (3 × 2 min, RT), and then stained with 0.5 µM HS-84 or HS-169, diluted in PBS, for 30 min at RT in the dark. After washing with PBS three times, the samples were mounted using Dako mounting medium for fluorescence (Agilent, Santa Clara, CA, USA). The result was analyzed using an inverted Zeiss 780 laser scanning confocal microscope (Zeiss, Oberkochen, Germany). The syntheses of HS-84 and HS-169 have been described elsewhere [37,38].

### Antibody and ligand triple staining

Protocol is described elsewhere [40]. Intestines were stained using anti-human Aβ antibody (Mabtech, Nacka Strand, Sweden) binding residues 3–8 of Aβ and 0.5 µM HS-84 and HS-169. No DAPI was used.

### Sample preparation for quantification studies

Offspring were snap-frozen in liquid nitrogen 8 days post eclosion. 20 female flies (Arctic-, $T_{22}$Aβ1−42-, and W1118 control flies) were decapitated and their bodies were homogenized in 60 µl extraction buffer [100 mM Hepes; 5 mM EDTA; 1X protease inhibitor cocktail]. The samples were centrifuged in 12 000 g at 10° for 5 min, and then all (soluble) supernatant was collected.* The pellet was homogenized in 40 µl extraction buffer + 5 M Gua-HCl. The homogenates were incubated for 10 min in RT, sonicated for 4 min, and then centrifuged as before. The (insoluble) supernatant was collected. The supernatants were diluted in 1:10 Diluent 35 (+ 0.5 M Gua-HCl for insoluble fractions). Aβ1−42 (6E10) calibration series (± 0.5 M Gua-HCl) were made according to the kit protocol (K151LBE-1, Meso Scale Discovery)

 * Additional two steps for the Gua-HCl gradient MSD analysis: The pellet was homogenized in 40 µl extraction buffer + 2.5 M Gua-HCl. The homogenates were incubated for 10 min in RT, centrifuged as before, and the (2.5 M fraction) supernatant was collected. This was done again but with 4 M Gua-HCl to receive the 4 M fraction.

### Quantification of Aβ species by MSD analysis

A V-PLEX Human Aβ1–42 Peptide (6E10) kit (K151LBE-1, Meso Scale Discovery) was used to measure levels of Aβ1–42 in Myo31DF-derived Aβ flies. The plate wells were blocked and incubated (1 h, RT, gentle agitation) in Diluent 35 and

then washed 3 times with PBS-tween before adding 50 μl calibrator (duplicate)/diluted sample (triplicate). The plate was incubated (1 h, RT, gentle agitation). The wells were washed as before and then incubated (1 h, RT, gentle agitation) with 25 μl 1X detection antibody (Sulfo tag 6E10, Meso Scale Discovery). After washing the wells as before, 150 μl 2X Reading buffer was added and the plate was analyzed using a MESO QuickPlex SQ 120MM instrument (Meso Scale Discovery).

## Statistical analysis

The data were analyzed using GraphPad Software 9.

## Supporting information

**S1 Fig. Several Aβ species detected by the antibody in T22Aβ1–42 flies are not detected by HS-84.** Detection of Aβ species in the anterior midgut of Drosophila flies expressing the tandem construct $T_{22}$Aβ1–42 in the enterocytes using the Myo31DF driver. The confocal microscope three-dimensional (3D) images show gut tissue costained with Mabtech anti-human Aβ antibody (magenta) and LCO ligand HS-84 (green) 8 days post eclosion. White circles: Aβ species labeled by the antibody and not HS-84. Scale bar, 40 μm.
(PNG)

**S2 Fig. Aβ species detected by the antibody, HS-84, and HS-169 in Arctic flies.** Detection of Aβ species in the anterior midgut of Drosophila flies expressing the Arctic mutant of Aβ1–42 in the enterocytes using the Myo31DF driver. The confocal microscope single-plane images show gut tissue costained with Mabtech anti-human Aβ antibody (magenta) and LCO ligands HS-84 (green) and HS-169 (orange) 8 days post eclosion. White arrowheads: Aβ species labeled by all three staining agents. Scale bar, 50 μm.
(PNG)

## Acknowledgments

We thank Damian Crowther for kindly providing Aβ flies and LiU core facilities.

## Author contributions

**Conceptualization:** Greta Elovsson, Therése Klingstedt, K Peter R Nilsson, Ann-Christin Brorsson.

**Funding acquisition:** Greta Elovsson, K Peter R Nilsson, Ann-Christin Brorsson.

**Investigation:** Greta Elovsson, Ann-Christin Brorsson.

**Methodology:** Greta Elovsson.

**Project administration:** Greta Elovsson.

**Resources:** K Peter R Nilsson.

**Supervision:** K Peter R Nilsson, Ann-Christin Brorsson.

**Validation:** K Peter R Nilsson.

**Visualization:** Greta Elovsson.

**Writing – original draft:** Greta Elovsson, Ann-Christin Brorsson.

**Writing – review & editing:** Greta Elovsson, Therése Klingstedt, K Peter R Nilsson, Ann-Christin Brorsson.

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
