## [Decision Letter · Decision Letter 0]

Dear Dr. Brorsson,

Thank you for submitting your manuscript to PLOS ONE. After careful consideration, we feel that it has merit but does not fully meet PLOS ONE’s publication criteria as it currently stands. Therefore, we invite you to submit a revised version of the manuscript that addresses the points raised during the review process.

**As presented in the appended comments below both reviewers express significant questions regarding technical aspects of the model itself questioning its validity. These concerns must be thoroughly addressed, with additional experiments as necessary. Clearly additional experiments are necessary to assess more than a single point aggregation assessment.**
**Both reviewers raise concerns regarding the use of lifespan as the single in vivo toxicity indicator and this concern must be thoroughly addressed.**
**Moreover, a number of additional concerns are raised regarding data presentation, statistical analyses and interpretation by both reviewers and these have to also be addressed point by point.**
**I recognise that these required changes including additional experiments will take time beyond the automatically allotted, so fell free to ask for additional time to complete these. **

We look forward to receiving your revised manuscript.

Kind regards,

Efthimios M. C. Skoulakis, PhD

Academic Editor

PLOS ONE

**Journal Requirements:**

Swedish Research Council (2016-00748)

Swedish Brain Foundation

Gun and Bertil Stohne’s Foundation

Åhléns Foundation (193059)

Swedish Alzheimer Foundation

4. Please amend your manuscript to include your abstract after the title page.

Reviewers' comments:

Reviewer's Responses to Questions

**Comments to the Author**

1. Is the manuscript technically sound, and do the data support the conclusions?

Reviewer #1: No

Reviewer #2: No

2. Has the statistical analysis been performed appropriately and rigorously?

Reviewer #1: No

Reviewer #2: No

3. Have the authors made all data underlying the findings in their manuscript fully available?

Reviewer #1: No

Reviewer #2: Yes

4. Is the manuscript presented in an intelligible fashion and written in standard English?

Reviewer #1: Yes

Reviewer #2: Yes

**Reviewer #1:**  This paper describes the characterisation of the aggregation state of two previously published Aß constructs, a wt double Aß and an arctic Aß. They describe different aggregation states for the two constructs and attempt to correlate this to their toxicity. The authors do all this in fly guts. Although fly guts are more accessible than neurons, and could potentially be used for biochemical characterisation of Aß aggregation, the authors should explain in more detail why this is a relevant or useful model. There are also some issues that need addressing before the manuscript is ready for publication.

1. It is not totally clear what model is used, do these Aß models contain a secretion model? Is it the same for the two? Are both peptides cleaved and secreted in this model or are the Aß peptides measured here intracellular?

2. Quantifications for the stainings are missing: HS-84 and HS-169 signals should be normalised and quantified with replicates. Co-localisation with Aß signal should also be quantified.

3. The authors claimed that they observed similar toxic effect (based on similar median survival days) and calculated the extent of reduction in lifespan per unit of Aβ. This was based on the assumption that the reduction in lifespan (i.e. toxicity) and Abeta levels are always linearly correlated without saturation effect. However, it is possible that Abeta levels (of a certain form or different forms combined) need to fall in a range to be correlated to the toxic outcomes, and further increasing its levels will not become more toxic? Using geneswitch systems at a gradient of RU doses may help determine whether the toxicity from Abeta is already saturated.

4. The authors base their conclusions on a single point for Aß aggregation. It is possible that the Arctic mutation leads to earlier aggregation, relative to the wt Aß constructs. The quantification of soluble/insoluble Aß should be performed at different timepoints, to show that they increase with similar dynamics.

4. The authors used lifespan as the only readout of Aβ toxicity. Do these gut models of Aβ have other phenotypic readouts, for example gut leakage?

5. The authors chose day 8 flies because “day 8 was near the median survival times for both genotypes”, but is Day8 really close to median survival (shown in Fig1A)?

6. Why is there so much more Aβ in the T22Aβ1-42 line than that in the Arctic line? Is it due to different transcription levels? Or degradation mechanisms clearing them at different rates? A qPCR showing transcription levels should be included to show whether transcription was equivalent.

7. Statistical analysis in Fig 4 or 5 is missing. This should be presented.

8 Expressing Aβ in neurons would be more relevant to disease context. Why did the authors focus on the gut? Authors should comment in more detail as to the usefulness of this model.

**Reviewer #2: ** This study investigated the heterogeneity of Aβ aggregates produced in the digestive tract in fruit flies expressing the Aβ1-42 peptide with the Arctic mutation E22G (Arctic fruit flies) or a dimeric construct of Aβ1-42 (T22Aβ1-42 fruit flies). Although fruit flies expressing these two Tg Aβ strains exhibited a similar degree of lifespan reduction, the Aβ produced by these flies showed different LCO staining patterns and differed in their solubility in Gua-HCl. Based on these results, the authors claim to have demonstrated that different Aβ aggregates have different pathological effects. It is interesting to note that the two fruit fly models they used had similar levels of toxicity despite different degrees of expression and aggregation. However, it is well known that the Arctic mutation is associated with increased aggregation and increased toxicity compared to wild-type Aβ. The experimental evidence presented in this study is unreliable and preliminary and does not sufficiently support their claims. Furthermore, some of the results appear to be a repetition or reuse of the authors' previous work. Here are the main points of criticism.

1) Alzheimer's disease is a neurodegenerative disease. The validity of the model of Aβ expression in the gut is questionable. It remains to be seen if the observations of this study are replicated in the brain.

2) Figure 1A appears to be a figure already used by the authors in a previously published paper (a).

a. Elovsson G, Klingstedt T, Brown M, Nilsson KPR, Brorsson AC. A Novel Drosophila Model of Alzheimer's Disease to Study Aβ Proteotoxicity in the Digestive Tract. Int J Mol Sci. 2024 Feb 9;25(4):2105. doi: 10.3390/ijms25042105.

3) Figure 1B, C, Figure 2, 3, etc. should be measured by quantitative methods. The single figure presented is insufficient to be reliable data. Comparison of more objects and more locations is needed.

4) In Figure 4, the amount of Aβ accumulated in T22Aβ1-42 fruit flies appears to be about 4 times greater than that in Arctic flies, but the differences in Figures 2 and 3 appear to be much greater.

5) Only one indicator of toxicity was presented, lifespan. However, lifespan is not a direct indicator of Aβ toxicity. Other parameters such as cell death should be used.

6) Lack of appropriate controls in Figures 2 and 3. It is questionable whether the two dyes used stained at the same brightness. The staining of the two dyes may be different depending on various factors such as staining method, confocal laser intensity, and post-experiment observation time. In Figure 2, we also observe green areas that stain with HS-84 but not with the antibody. How to explain this is questionable.

**Do you want your identity to be public for this peer review?** For information about this choice, including consent withdrawal, please see our Privacy Policy

Reviewer #1: No

Reviewer #2: No

---

## [Author Response · Author response to Decision Letter 1]

19 Mar 2025

Dear reviewers,

See the attached file, Author response to reviewers

---

## [Decision Letter · Decision Letter 1]

Dear Dr. Brorsson,

Thank you for submitting your manuscript to PLOS ONE. After careful consideration, we feel that it has merit but does not fully meet PLOS ONE’s publication criteria as it currently stands. Therefore, we invite you to submit a revised version of the manuscript that addresses the points raised during the review process.

We look forward to receiving your revised manuscript.

Kind regards,

Efthimios M. C. Skoulakis, PhD

Academic Editor

PLOS ONE

Journal Requirements:

Additional Editor Comments:

**Although improved, please see reviewer comments for additional changes. Please make sure you discuss **
**extensively how your current work differs and possibly adds to prior similar publications**

Reviewers' comments:

Reviewer's Responses to Questions

**Comments to the Author**

Reviewer #1: (No Response)

Reviewer #2: (No Response)

2. Is the manuscript technically sound, and do the data support the conclusions?

Reviewer #1: Partly

Reviewer #2: No

3. Has the statistical analysis been performed appropriately and rigorously?

Reviewer #1: No

Reviewer #2: No

4. Have the authors made all data underlying the findings in their manuscript fully available?

Reviewer #1: Yes

Reviewer #2: Yes

5. Is the manuscript presented in an intelligible fashion and written in standard English?

Reviewer #1: Yes

Reviewer #2: Yes

Reviewer #1: The authors have addressed some of our concerns. A few points still need addressing:

-there is no Fig 4B in the manuscript that I could find

-there is no statistical analysis for the guts phenotype

-the gut phenotype is not discussed, if there is a significant difference, what does this mean for toxicity.

-the conclusions should be toned down, as the authors have not been able to show that the toxicity does not have a thresholding effect and anything higher than a certain amount of Aß will always give the same toxicity, they should add a sentence to this effect in the discussion

Reviewer #2: PONE-D-24-52919R1

Diversity of Aβ aggregates produced in a gut-based Drosophila model of Alzheimer’s disease

Elovsson et al.

There remain concerns regarding the novelty of this study. As pointed out in the previous review, it is well established that the Arctic mutation is associated with increased aggregation and enhanced toxicity compared to wild-type Aβ (Iijima et al., 2008; Perálvarez-Marín et al., 2009). Moreover, the exact same model expressing these Aβ in the gut has already been published by the current authors themselves (Elovsson et al., 2024). It is therefore unclear what additional advancement this manuscript offers beyond what has already been reported.

Perálvarez-Marín A, Mateos L, Zhang C, Singh S, Cedazo-Mínguez A, Visa N, Morozova-Roche L, Gräslund A, Barth A. Influence of residue 22 on the folding, aggregation profile, and toxicity of the Alzheimer's amyloid beta peptide. Biophys J. 2009 Jul 8;97(1):277-85.

Iijima K, Chiang HC, Hearn SA, Hakker I, Gatt A, Shenton C, Granger L, Leung A, Iijima-Ando K, Zhong Y. Abeta42 mutants with different aggregation profiles induce distinct pathologies in Drosophila. PLoS One. 2008 Feb 27;3(2):e1703. doi: 10.1371/journal.pone.0001703. PMID: 18301778; PMCID: PMC2250771.

Elovsson G, Klingstedt T, Brown M, Nilsson KPR, Brorsson AC. A Novel Drosophila Model of Alzheimer's Disease to Study Aβ Proteotoxicity in the Digestive Tract. Int J Mol Sci. 2024 Feb 9;25(4):2105. doi: 10.3390/ijms25042105. PMID: 38396782; PMCID: PMC10888607.

Minor point

Statistical error in Figure 6A: The result does not appear to be non-significant (ns) as indicated.

**Do you want your identity to be public for this peer review?** For information about this choice, including consent withdrawal, please see our Privacy Policy

Reviewer #1: No

Reviewer #2: No

---

## [Author Response · Author response to Decision Letter 2]

19 Jun 2025

Reviewer #1: The authors have addressed some of our concerns. A few points still need addressing:

-there is no Fig 4B in the manuscript that I could find

Author response: Yes – that is correct, there is only one Figure 4 that is referred to in the manuscript and that has been resubmitted.

-there is no statistical analysis for the guts phenotype

Author response: Statistical analysis of the gut phenotype yielded a P-value greater than 0.05, which strongly suggests that the result is likely under the null hypothesis. Although this intriguing phenotype was observed in Arctic flies 8 days post-eclosion, it is not statistically significant. Therefore, in the paper, we cautiously interpret this finding as a potential indication of a toxic mechanism in Arctic flies that differs from the toxic events in the T22Aβ1-42 flies.

-the gut phenotype is not discussed, if there is a significant difference, what does this mean for toxicity.

Author response: See the answer above

-the conclusions should be toned down, as the authors have not been able to show that the toxicity does not have a thresholding effect and anything higher than a certain amount of Aß will always give the same toxicity, they should add a sentence to this effect in the discussion

Author response: The discussion has been updated at line 378-384. “Although the survival data revealed a similar reduction in median lifespan between the two fly genotypes, a potential saturation effect of Aβ accumulation should be considered which could mean that once Aβ levels reach a certain threshold, additional accumulation does not lead to increased toxicity. However, the outcome from the calculation of the proteotoxic effect showed that the Aβ1-42 pool produced in Arctic flies was 4 times more proteotoxic than the Aβ1-42 pool produced in T22Aβ1-42 flies revealing that the toxic effect per Aβ unit (pg/fly) is higher in the Arctic flies compared to the T22Aβ1-42 flies.”

Reviewer #2: PONE-D-24-52919R1

Diversity of Aβ aggregates produced in a gut-based Drosophila model of Alzheimer’s disease

Elovsson et al.

There remain concerns regarding the novelty of this study. As pointed out in the previous review, it is well established that the Arctic mutation is associated with increased aggregation and enhanced toxicity compared to wild-type Aβ (Iijima et al., 2008; Perálvarez-Marín et al., 2009). Moreover, the exact same model expressing these Aβ in the gut has already been published by the current authors themselves (Elovsson et al., 2024). It is therefore unclear what additional advancement this manuscript offers beyond what has already been reported.

Author response: The novelty of this study lies in the detailed in vivo characterization of Aβ aggregates formed in Arctic flies and in flies expressing a dimeric construct of Aβ1–42. Specifically, we demonstrate how these aggregates can be analyzed for their morphological properties using luminescent conjugated oligothiophenes (LCOs), which selectively bind to Aβ aggregates. Furthermore, we reveal distinct differences in the in vivo aggregation behavior of the two Aβ variants: the T22Aβ1–42 peptide tends to form aggregates that are largely insoluble in 5 M guanidine hydrochloride (Gua), whereas the Arctic peptide forms aggregates that are more readily solubilized at lower Gua concentrations. These findings provide valuable insights into the aggregation dynamics of Aβ peptides and offer a foundation for future studies aimed at elucidating the relationship between Aβ aggregation and its toxic mechanisms, ultimately contributing to a better understanding of Alzheimer’s disease pathogenesis.

Perálvarez-Marín A, Mateos L, Zhang C, Singh S, Cedazo-Mínguez A, Visa N, Morozova-Roche L, Gräslund A, Barth A. Influence of residue 22 on the folding, aggregation profile, and toxicity of the Alzheimer's amyloid beta peptide. Biophys J. 2009 Jul 8;97(1):277-85.

Iijima K, Chiang HC, Hearn SA, Hakker I, Gatt A, Shenton C, Granger L, Leung A, Iijima-Ando K, Zhong Y. Abeta42 mutants with different aggregation profiles induce distinct pathologies in Drosophila. PLoS One. 2008 Feb 27;3(2):e1703. doi: 10.1371/journal.pone.0001703. PMID: 18301778; PMCID: PMC2250771.

Elovsson G, Klingstedt T, Brown M, Nilsson KPR, Brorsson AC. A Novel Drosophila Model of Alzheimer's Disease to Study Aβ Proteotoxicity in the Digestive Tract. Int J Mol Sci. 2024 Feb 9;25(4):2105. doi: 10.3390/ijms25042105. PMID: 38396782; PMCID: PMC10888607.

Minor point

Statistical error in Figure 6A: The result does not appear to be non-significant (ns) as indicated.

Author response: This result is not statistically significant, likely due to the extremely low levels of Aβ detected (below 1 pg per fly) which are near the lower limit of the assay’s sensitivity.

---

## [Editor Report · Decision Letter 2]

Diversity of Aβ aggregates produced in a gut-based Drosophila model of Alzheimer’s disease

PONE-D-24-52919R2

Dear Dr. Brorsson,

We’re pleased to inform you that your manuscript has been judged scientifically suitable for publication and will be formally accepted for publication once it meets all outstanding technical requirements.

Kind regards,

Efthimios M. C. Skoulakis, PhD

Academic Editor

PLOS ONE
---

## [Editor Report · Acceptance letter]

PONE-D-24-52919R2

PLOS ONE

Dear Dr. Brorsson,

I'm pleased to inform you that your manuscript has been deemed suitable for publication in PLOS ONE. Congratulations! Your manuscript is now being handed over to our production team.

Kind regards,

on behalf of

Dr. Efthimios M. C. Skoulakis

Academic Editor

PLOS ONE